# Exact Learning Dynamics of a Linear Autoencoder Through Diagram Expansions

**Eugene Golikov, Yaroslav Gusev**
Applied AI Institute
Moscow, Russia
e.golikov@applied-ai.ru

**Dmitry Yarotsky**
Applied AI Institute & Steklov Institute of Mathematics
Moscow, Russia

## Abstract

We consider the problem of factorizing the identity matrix as a product of two matrices - that is, learning a shallow linear autoencoder - with gradient flow. We formally expand the loss as a function of time, and show how the expansion terms could be expressed via graphs akin to Feynman diagrams. This turns the problem of computing the loss as a function of training time into a purely combinatorial problem. By analyzing this problem, we provide a complete classification of learning regimes in the limit of large input and hidden size. These limit regimes can be identified with various faces of a *Pareto polygon* describing the problem. We compute loss expansion terms exactly in the limit regimes corresponding to most of these faces. These formal expansions turn out to be summable and lead to analytic formulas for the average loss evolution which agree very well with the experiment. We emphasize that some of our solutions correspond to intrinsically nonlinear training dynamics; such solutions are scarce in the literature.

## 1 Introduction

**Motivation.** One of the major theoretical challenges in constructing a theory of neural networks stems from the fact that integrating their training dynamics is a notoriously difficult problem. This is because the maps implemented by neural networks are highly nonlinear. One may hope that the training dynamics may simplify in a suitable "macroscopic" limit in which the input/output size of the target and/or the number of hidden units (width) of the model are large. Indeed, in the well-known NTK regime (Jacot et al., 2018) the process of training a neural network becomes equivalent to a kernel method and is thus mathematically tractable.

However, while being a very convenient model for a theorist, kernel methods train linearly and do not learn features, while neural networks train non-linearly, hence do learn features — this is believed to be a source of superior performance of the latter (Chizat et al., 2019; Fort et al., 2020). In fact, large-dimension limits that exhibit non-linear training do exist (e.g. mean-field (Rotskoff & Vanden-Eijnden, 2018; Chizat & Bach, 2018; Sirignano & Spiliopoulos, 2020) and $\mu P$ limits (Yang & Hu, 2021)). However, a complete classification of large-dimension limits is still lacking. Moreover, while the dynamics is easy to integrate in the kernel limit, exact solutions in non-linear regimes are scarce in the literature. See further discussion of related work in Appendix A.3.

An *autoencoder* is a neural network trained to recover its own input. Such models date back to Rumelhart et al. (1985) and have a number of applications including feature extraction (Erhan et al., 2010), denoising and anomaly detection (Vincent et al., 2008), and dimensionality reduction (Hinton & Salakhutdinov, 2006); see Appendix A.2 for a short overview. In the present work, we consider a simple problem of learning a linear autoencoder with one hidden layer. While being equivalent to simple linear models, linear networks exhibit non-trivial training dynamics (Saxe et al., 2013); see Appendix A.1 for an overview. Our setup is a particular case of the identity tensor decomposition problem considered in Yarotsky et al. (2026) for $\nu = 2$ in the asymmetric scenario. According to the classification of limit learning regimes presented in Yarotsky et al. (2026), all these regimes could be associated either with vertices, or with sides of a *Pareto-polygon*, or with the whole polygon itself. We present exact solutions for most of the faces of the Pareto-polygon. Some of these faces we were able to solve exhibit nonlinear training dynamics.

**Our contribution and the structure of the paper.**

1. We consider a problem of learning a linear autoencoder using a quadratic loss. Following Yarotsky et al. (2026), we formally expand the loss as a power series wrt training time and compute the limit of each term using specific diagram calculus. We briefly describe this method in Section 3.

2. The method of Yarotsky et al. (2026) leads to a complete classification of limit regimes: Theorem 1 in Section 4. Each limit regime could be associated with a face of a *Pareto polygon* Fig. 2; see Section 5 for their natural interpretation.

3. In most of these regimes, the training dynamics could be integrated either via directly computing the loss expansion terms or, more elegantly, via a method of characteristics. In Section 6, we present explicit solutions for the learning dynamics corresponding to all faces of the Pareto-polygon except for one. We note that Edge BC is the only edge corresponding to linearized training, the rest exhibit nonlinear training dynamics. The obtained analytic solutions match very well with experiment: see Fig. 3.

## 2  THE SETTING: GF FOR IDENTITY MATRIX DECOMPOSITION

We call a *linear autoencoder* a model of the following form.

$$\boldsymbol{f}(\boldsymbol{x}) = U^\top V \boldsymbol{x}, \tag{1}$$

where $\boldsymbol{x} \in \mathbb{R}^p$, while $U, V \in \mathbb{R}^{H \times p}$. Training it corresponds to minimizing the following loss function.

$$L_X(U, V) = \frac{1}{2N} \left\| U^\top V X - X \right\|_F^2, \tag{2}$$

where $X \in \mathbb{R}^{p \times N}$ encodes a dataset of $N$ samples. As long as $H \geq p$, the optimal values of $U$ and $V$ satisfy $U^\top V = I$.

When each data sample comes from a distribution with zero mean and identity covariance, the above loss function converges as $N \to \infty$ by Law of Large Numbers:

$$L_X(U, V) \xrightarrow{\text{a.s.}} \frac{1}{2} \mathbb{E}_{\boldsymbol{x}} \left\| U^\top V \boldsymbol{x} - \boldsymbol{x} \right\|_2^2 = \frac{1}{2} \left\| U^\top V - I \right\|_F^2 \overset{\text{def}}{=} L(U, V). \tag{3}$$

We consider optimizing the infinite-data loss $L$ with gradient flow with learning rate $1/T$:

$$\frac{dU(t)}{dt} = -\frac{1}{T} \frac{\partial L(U(t), V(t))}{\partial U}, \qquad \frac{dV(t)}{dt} = -\frac{1}{T} \frac{\partial L(U(t), V(t))}{\partial V}, \tag{4}$$

and the following initial conditions.

$$U_{i,k}(0) \sim \mathcal{N}(0, \sigma^2), \qquad V_{i,k}(0) \sim \mathcal{N}(0, \sigma^2), \qquad \forall i \in [H] \quad \forall k \in [p]. \tag{5}$$

This setting corresponds to canonical polyadic identity tensor decomposition studied in Yarotsky et al. (2026), in the $\nu = 2$ asymmetric case. We briefly describe their method, particularly applied to our case, in the following sections.

## 3  LOSS EVOLUTION AND DIAGRAM CALCULUS

### 3.1  DIAGRAMMATIC REPRESENTATION

The loss function we optimize could be expressed as a sum of three terms:

$$L(U, V) = \frac{1}{2} \left\| U^\top V - I \right\|_F^2 = \frac{1}{2} \left\| U^\top V \right\|_F^2 - \text{Tr} \left[ U^\top V \right] + \frac{p}{2}. \tag{6}$$

The first term encodes the interaction of the model with itself. Let us expand it in terms of sums:

$$\left\| U^\top V \right\|_F^2 = \sum_{i,j=1}^p \sum_{k=1}^H \sum_{k'=1}^H U_{k,i} V_{k,j} U_{k',i} V_{k',j}. \tag{7}$$

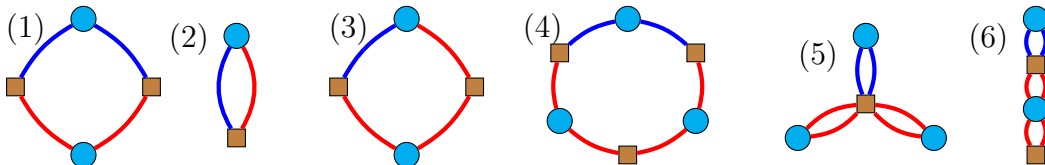

Figure 1: Various diagram appearing in the study of linear autoencoder (1). Blue circles denote $p$-nodes, while brown squares denote $H$-nodes. Blue edges correspond to the weight matrix $U$, while red ones — to $V$. Left to right: (1) Diagram $D$; (2) Diagram $R$; (3) One of the diagrams resulting from merging $D$ and $R$: there are two such diagrams, and two diagrams resulting from inverting the colors; (4) One of the diagrams resulting from merging $D$ and $D$: similarly, there are four such diagrams, and four diagrams resulting from inverting the colors; (5), (6) Two possible valid contractions of the diagram (4).

Expressions like this could be conveniently associated with a graph defined as follows.

Let us associate a vertex to every summation index, and an edge connecting two vertices — to a matrix entry with the corresponding indices. We shall refer to vertices associated with summations over $p$ terms (like $i$ and $j$ in the above formula) as *p-vertices*, while to that associated with summations over $H$ terms (like $k$ and $k'$) as *H-vertices*. We shall also distinguish edges associated with different matrices ($U$ or $V$) by their color. This results in the leftmost graph of Fig. 1. We shall denote this diagram $D$, and use the diagram and the corresponding expression interchangeably in the following.

Expanding the second term gives

$$\mathrm{Tr}\left[U^\top V\right] = \sum_{i=1}^{p}\sum_{k=1}^{H} U_{k,i}V_{k,i}. \tag{8}$$

The corresponding graph is shown on Fig. 1 second; we shall denote it with $R$.

The last term of Eq. (6) could be associated with a trivial empty graph. This way, the whole loss function is associated with a linear combination of graphs of the form described above:

$$L = \frac{1}{2}D - R + \frac{p}{2}. \tag{9}$$

## 3.2 Loss expansion

We are interested in the loss evolution under gradient flow, Eq. (4). Let us formally expand the loss as a function of time $t$:

$$L(t) = \sum_{s=0}^{\infty} \frac{d^s L}{dt^s}(0)\frac{t^s}{s!}. \tag{10}$$

As we shall demonstrate shortly, the higher-order derivatives wrt $t$ could be conveniently expressed in terms of *diagram merging*.

**Derivatives and diagram merging.** Indeed, let us consider a scalar quantity $G$ that depends on the learned weights $U$ and $V$. It evolves under gradient flow as follows.

$$\frac{dG}{dt} = -\frac{1}{T}\sum_{i=1}^{p}\sum_{k=1}^{H}\left(\frac{\partial L}{\partial U_{k,i}}\frac{\partial G}{\partial U_{k,i}} + \frac{\partial L}{\partial V_{k,i}}\frac{\partial G}{\partial V_{k,i}}\right). \tag{11}$$

In particular, this allows us to compute the loss derivatives of any order recursively:

$$\frac{d^s L}{dt^s} = -\frac{1}{T}\sum_{i=1}^{p}\sum_{k=1}^{H}\left(\frac{\partial L}{\partial U_{k,i}}\frac{\partial}{\partial U_{k,i}}\frac{d^{s-1}L}{dt^{s-1}} + \frac{\partial L}{\partial V_{k,i}}\frac{\partial}{\partial V_{k,i}}\frac{d^{s-1}L}{dt^{s-1}}\right) \qquad \forall s \in \mathbb{N}. \tag{12}$$

If $G$ is represented as a *valid diagram*, i.e. as a simple graph consisting of p- and H-vertices, and blue and red edges, such that every edge connects vertices of different types, then $\frac{dG}{dt}$ could be associated with a linear combination of valid diagrams via the following *merging* procedure.

We pick an edge from $G$ together with an edge of the same color from $L$. We then remove both edges and identify the corresponding vertices of matching types. Doing this for all pairs of same-color edges yields a sum of graphs, we shall denote it with $G \star L$. The reader could easily notice that Eq. (11) reads precisely as

$$\frac{dG}{dt} = -\frac{1}{T} G \star L. \tag{13}$$

Note that the merger operation is commutative but *not* associative. The two middle pictures of Fig. 1 show diagrams from $D \star D$ and $D \star R$. This way, the loss evolution Eq. (10) could be expressed as

$$L(t) = \frac{p}{2} + \sum_{s=0}^{\infty} \left( \frac{1}{2} D(0) - R(0) \right)^{\star(s+1)} \frac{(-t)^s}{s! T^s}. \tag{14}$$

Here, every summation term is a finite linear combination of valid diagrams. We shall drop argument 0 in $D(0)$ and $R(0)$ in the sequel as we will not encounter any other arguments.

It is easy to see that in our problem setting, $\left( \frac{1}{2} D(0) - R(0) \right)^{\star(s+1)}$ is always a linear combination of loops with alternating vertex types, such that every p-vertex has adjacent edges of the same color, while every H-vertex has adjacent edges of different colors. We emphasize, however, that this is not the case for more complex loss functions, such as ones considered in Yarotsky et al. (2026).

### 3.3 Averaging over random weight initialization

Eq. (14) yields a random variable that is still difficult to analyze. Let us consider its expectation over random weight initialization Eq. (5). Following Yarotsky et al. (2026), we swap the expectation and the infinite sum without providing any rigorous justification:

$$\boxed{\mathbb{E}[L(t)] = \frac{p}{2} + \sum_{s=0}^{\infty} \mathbb{E}\left[ \left( \frac{1}{2} D - R \right)^{\star(s+1)} \right] \frac{(-t)^s}{s! T^s}.} \tag{15}$$

Taking the expectation of an expression associated to a valid diagram could be done via *diagram contraction*.

**Averaging and diagram contraction.** As described in Eq. (5), every weight is Gaussian with zero mean at $t = 0$. Therefore, the expectation of every polynomial in initial weights could be computed using Wick's formula. Every valid diagram is associated with a sum of products of weights, hence it is indeed a polynomial. As described by Wick's formula, the expectation of a valid diagram $G$ is a sum of terms indexed by all partitions of edges into pairs. For each such pair, $\mathbb{E}[U_{k,i} U_{k',i'}] = \mathbb{E}[V_{k,i} V_{k',i'}] = \sigma^2 \delta_{i=i'} \delta_{k=k'}$, while $\mathbb{E}[U_{k,i} V_{k',i'}] = 0$. As a result, the contribution of a partition to $\mathbb{E}[G]$ equals a deterministic combinatorial expression associated with a contracted diagram. We show valid (i.e. yielding non-zero expectation) contractions in the two rightmost diagrams of Fig. 1.

Computing $\mathbb{E}\left[ \left( \frac{1}{2} D - R \right)^{\star(s+1)} \right]$ therefore becomes a combinatorial problem of counting all possible valid contractions of $\left( \frac{1}{2} D - R \right)^{\star(s+1)}$ for every $s$. This expression is always polynomial in $p$, $H$, and $\sigma^2$:

$$Y_s \overset{\text{def}}{=} \mathbb{E}\left[ \left( \frac{1}{2} D - R \right)^{\star(s+1)} \right] = \sum_{q=1}^{s+2} \sum_{n=1}^{s+2} \sum_{l=1}^{s+2} c_{q,n,l;s} p^q H^n \sigma^{2l}. \tag{16}$$

Indeed, a valid contraction is still a valid diagram. Each edge pair contributes a factor of $\sigma^2$, while every vertex yields a factor of $p$ or $H$, depending on its type. Since merging with $D$ adds one of each type, while merging with $R$ does not add any, and contraction always leaves at least one vertex of each type, $q, n \in [s+2]$ Similarly, merging with $D$ adds two edges, while merging with $R$ does not add any; this gives $l \in [s+2]$.

## 4 Large scale limit

Evaluating the expression for the loss expansion coefficients, Eq. (16), still yields complicated combinatorics. Fortunately, the problem simplifies as $p$ and $H$ go to infinity: in this case, some of the

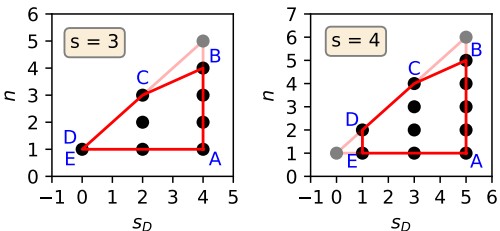

Figure 2: Pareto optimal terms (black dots; see Theorem 1) and the corresponding Pareto polygons (red).

Table 1: Faces of the Pareto polygon of Fig. 2 along with their interpretations.

| Face | Scaling condition | Natural $T$ | Parameterization | Learning | Interpretation |
|------|-------------------|-------------|-------------------|----------|----------------|
| A – B | $H \asymp p, p\sigma^2 \to \infty$ | $H\sigma^2$ | Balanced | No | Free evolution |
| B – C | $pH\sigma^4 \asymp 1, H\sigma^2 \to \infty$ | $H\sigma^2$ | Over- | Lazy | NTK |
| C | $pH\sigma^4 \to 0, H\sigma^2 \to \infty$ | $H\sigma^2$ | Over- | Lazy | NTK, $f(0) \equiv 0$ |
| C – D | $H\sigma^2 \asymp 1, H/p \to \infty$ | $1$ | Over- | Rich | Mean-field |
| D – E | $H \asymp p, H\sigma^2 \to 0$ | $1$ | Balanced | Rich | — |
| E – A | $p\sigma^2 \asymp 1, H/p \to 0$ | $1$ | Under- | Rich | — |

monomials dominate the others. Taking into account only these dominating terms greatly simplifies the calculations. It turns out that such formal series, where only dominating monomials are taken into account, can be summed explicitly and match very well with the experiment (see Fig. 3).

Following Yarotsky et al. (2026), we consider power-law scaling:

$$p \asymp a^{\alpha_p}, H \asymp a^{\alpha_H}, \sigma \asymp a^{\alpha_\sigma}, \quad a \to +\infty, \tag{17}$$

with specific exponents $\boldsymbol{\alpha} = (\alpha_p, \alpha_H, \alpha_\sigma)$. Then the polynomial $Y_s$ defined in Eq. (16) behaves as

$$Y_s \asymp a^{\alpha_{Y_s}}, \qquad \alpha_{Y_s} = \max_{(q,n,l):\, c_{q,n,l;s} \neq 0} (\alpha_p q + \alpha_H n + 2\alpha_\sigma l). \tag{18}$$

We call the monomials yielding the above maximum *leading*. As shown in Yarotsky et al. (2026), all leading terms are necessarily *Pareto-optimal* ones. We call a monomial $p^q H^n \sigma^{2l}$ Pareto-optimal if there is no other monomial $p^{q'} H^{n'} \sigma^{2l}$ in $Y_s$ such that $q' \geq q$ and $n' \geq n$.

Pareto-optimal monomials in $Y_s$ correspond to *minimally-contracted* diagrams, i.e. diagrams and pairings that minimally constrain summations over node indices. Since all diagrams we encounter in our setting are loops, minimally-contracted ones are trees with double edges, see the last two diagrams of Fig. 1. This property greatly simplifies calculations.

We shall look for leading terms among the Pareto-optimal ones. Remarkably, all Pareto-optimal terms belong to a common plane in a three-dimensional space of $(q, n, 2l)$. This claim follows from the complete description of Pareto-optimal terms below:

**Theorem 1** (Yarotsky et al. (2026)). *Up to nonzero factors, the Pareto-optimal terms in $Y_s$ are*

$$p^{s_D + 2 - n} H^n \sigma^{2(s_D + 1)}, \tag{19}$$

*where $0 \leq s_D \leq s + 1$ such that $s_R = s + 1 - s_D$ is even, and $1 \leq n \leq s_D + 1$. We also have to exclude the term $(n, s_D) = (s + 2, s + 1)$.*

These terms are depicted with black dots in Fig. 2. Here, gray dots denote terms we had to exclude.

## 5 PARETO POLYGONS AND LEARNING REGIMES

Since there is a finite number of Pareto-optimal terms for a given $s$, its convex hull is a polygon that Yarotsky et al. (2026) call *Pareto polygon*. It is a pentagon ABCDE for even $s$; as points D and E coincide for odd $s$, it becomes a quadrangle in this case, see Fig. 2.

Since the Pareto-optimal terms lie on a common plane, there exists a scaling $\boldsymbol{\alpha} = (1, 1\frac{1}{2})$ that makes all Pareto-optimal terms leading. Any other scaling results in only a subset of the Pareto-terms

dominating. Since a scaling $\alpha$ acts linearly on powers $(q, n, l)$ to give a scaling of each monomial, as described in Eq. (18), the leading terms have to belong to faces of the Pareto polygon, i.e. to edges or vertices. Table 1 summarizes some of the corresponding scaling conditions together with their interpretations. It also lists natural scaling of the inverse learning rate $T$ obtained by balancing the base of power $s$ in the terms listed in Theorem 1 with $T$. We briefly reproduce the interpretations for different learning regimes below. For a more detailed overview, we refer the reader to Yarotsky et al. (2026).

As one moves bottom-up, $n$ increases, hence the corresponding limit regime becomes more and more parameterized. That is, all points on EA correspond to underparameterized regimes, while that on BC and CD — to overparameterized ones. Finally, AB and DE are called *balanced*: they require $p \asymp H$ to realize.

As one moves left to right in Fig. 2, $s_D$ increases, while $s_R$ increases. The rightmost edge AB corresponds to $s_R = 0$, which means that no $R$ diagrams are used in computing $Y_s$. Since $R$ diagrams encode model-target interaction, the model evolves ignoring the target in the corresponding limit regimes. Yarotsky et al. (2026) call this regime *free evolution*. We provide explicit solutions for Points A and B in Section 6.1.

As one moves left, model-target interaction starts to emerge. C is the only extreme point that corresponds to minimal model-target interactions among non-free regimes. One can associate this regime (and also that of Edge BC) with *linearized training*, or to the *NTK limit* of Jacot et al. (2018). We provide explicit solutions for Point C and Edge BC in Section 6.2. The loss falls exponentially, which indicates linear dynamics in the parameter space.

Moving further left yields regimes with richer model-target interactions. As such, Edge CD corresponds to the well-known *mean-field* limit of two-layer nets (Rotskoff & Vanden-Eijnden, 2018; Chizat & Bach, 2018; Sirignano & Spiliopoulos, 2020). There is also an underparameterized counterpart of the mean-field limit: Edge EA. The dynamics for both of these regimes is no longer linear; we provide explicit solutions in Sections 6.3 and 6.4. As the reader can notice, the loss no longer falls exponentially. Similarly, Edge DE corresponds to a *maximally interacting regime*; we provide the corresponding explicit solution in Section 6.5.

# 6 EXACT SOLUTIONS

In this section, we present explicit solutions for most of the faces of the Pareto-polygon. Some of them (A, B, C, D, E, BC, DE) allow for direct calculation of the leading terms of $Y_s$, while CD and AE require a more advanced technique based on the method of characteristics. We refer the reader to Section 6 of Yarotsky et al. (2026) for further details on this method.

## 6.1 FREE EVOLUTION: EDGE AB

Here we have only solutions for extreme points.

**Point A.** This corresponds to $s_R = 0$ and $n = 1$. This means that the model does not interact with the target in the limit, and we have to keep only one $H$-node after contraction. Therefore, all $H$-nodes have to be contracted into one thus forming a "flower" akin to one of the second to the last diagram of Fig. 1.

Point A is Pareto maximal when it dominates its neighbor points in the convex span, i.e. when $p^{s+2}H\sigma^{2s+4} \gg p^{s+1}H^2\sigma^{2s+4}$ and $p^{s+2}H\sigma^{2s+4} \gg p^s H\sigma^{2s}$. This is equivalent to $p \gg H$ and $p\sigma^2 \gg 1$. The following is a consequence of Eq. (15) and Proposition 1 in Appendix B.

$$\mathbb{E}\left[L(t)\right] \sim \frac{p}{2} + \frac{p^2 H \sigma^4}{2\left(1 + \frac{2p\sigma^2 t}{T}\right)^2}, \qquad H \to \infty, \quad p/H \to \infty, \quad p\sigma^2 \to \infty. \tag{20}$$

The first term is the loss of a zero model, while the second one is the loss of the free evolution. No target learning occurs as there is no interaction with the target.

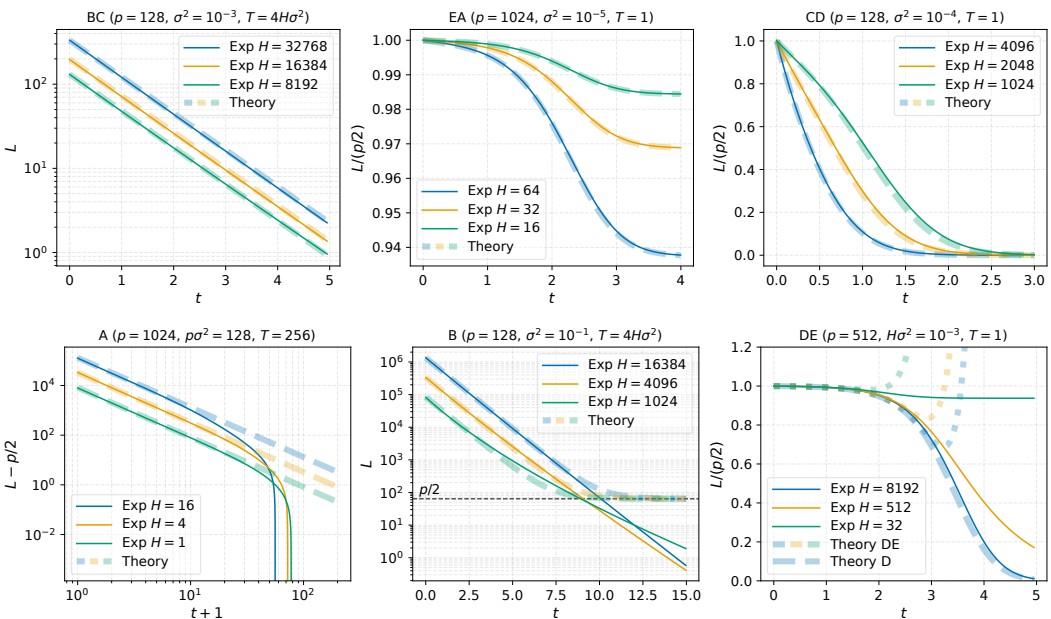

Figure 3: Experimental confirmation of the exact solutions from Sec. 6. **Top row.** *Left:* Eq. (23), *Middle:* Eq. (29), *Right:* Eq. (30). **Bottom row.** *Left:* Eq. (20), *Middle:* Eq. (21), *Right:* Eq. (31) (dotted thick lines) and Eq. (33) (dashed thick blue line).

See Fig. 3, bottom left, for an experimental validation. The formula matches very well with the experiment up to some $t$. After that, since both $H$ and $p$ are finite, the target gets learned (as indicated by a sudden drop of solid curves), while it never gets learned in the "free" limit considered.

**Point B.** This corresponds to $s_R = 0$ and $n = s + 1$. This model also does not interact with the target in the limit, but in contrast to Point A, and contrary to intuition, we have to keep two $p$-node after contraction. This is because $H$-nodes that would become petal ends of a flower if we contracted all $p$ nodes into one, are all "trans-nodes", i.e. there is a change of color at them. Hence such a contraction cannot be valid (it gives zero expectation in Wick's formula). This gives a diagram akin to one of the rightmost diagram of Fig. 1.

Point B is Pareto maximal when it dominates its neighbor points in the convex span, i.e. when $p^2 H^{s+1} \sigma^{2s+4} \gg p^3 H^s \sigma^{2s+4}$ and $p^2 H^{s+1} \sigma^{2s+4} \gg p H^s \sigma^{2s}$. This is equivalent to $H \gg p$ and $pH\sigma^4 \gg 1$. The following is a consequence of Eq. (15) and Proposition 2 in Appendix B.

$$\mathbb{E}\left[L(t)\right] \sim \frac{p}{2} + \frac{p^2 H \sigma^4}{2} e^{-\frac{4H\sigma^2 t}{T}}, \qquad p \to \infty, \quad H/p \to \infty, \quad pH\sigma^4 \to \infty. \tag{21}$$

Though $pH\sigma^4 \gg 1$, we kept the first term to underline that the target is never learned. See Fig. 3, bottom center, for an experimental validation. The same remark as for Point A applies here.

### 6.2 Linearized training: Edge BC

**Point C.** This corresponds to $s_R = 2$ and $n = s$. Point C is Pareto maximal when it dominates its neighbor points in the convex span, i.e. when $pH^s \sigma^{2s} \gg pH^{s-2}\sigma^{2s-4}$ and $pH^s \sigma^{2s} \gg p^2 H^{s+1} \sigma^{2s+4}$. This is equivalent to $H\sigma^2 \gg 1$ and $pH\sigma^4 \ll 1$. The following is a consequence of Eq. (15) and Proposition 3 in Appendix B.

$$\mathbb{E}\left[L(t)\right] \sim \frac{p}{2} e^{-\frac{4H\sigma^2 t}{T}}, \qquad p, H \to \infty, \quad H\sigma^2 \to \infty, \quad pH\sigma^4 \to 0. \tag{22}$$

The average loss equals $\frac{p}{2}$, the loss of a zero model, at $t = 0$. Therefore, one could think that in this limit, the model starts evolving from the zero one.

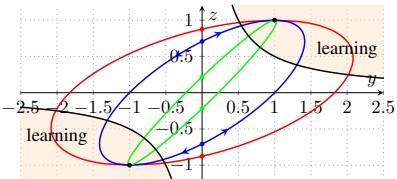

Figure 4: Elliptic characteristics of PDE (27) in the $yz$ plane. "Learning" indicates zones where the loss drops below the loss $\frac{p}{2}$ of the trivial zero model.

**Edge BC.** Points B and C have to be of the same order: $p^2 H^{s+1}\sigma^{2s+4} \approx pH^s\sigma^{2s}$, i.e. $pH\sigma^4 \approx 1$, or $\sigma^2 \approx 1/\sqrt{pH}$, which is NTK scaling (see Jacot et al. (2018)). Edge BC is Pareto maximal when it dominates its neighbor points in the convex span, i.e. when $p^2 H^{s+1}\sigma^{2s+4} \gg p^3 H^s\sigma^{2s+4}$ and $pH^s\sigma^{2s} \gg pH^{s-2}\sigma^{2s-4}$. This is equivalent to $H \gg p$ and $H\sigma^2 \gg 1$. Since $\sigma^2 \asymp 1/\sqrt{pH}$, the second condition is redundant, and we are left with $H \gg p$.

Since Edge BC consists of only two Pareto-optimal points for any $s$, B and C, while these two regimes do not have any common optimal diagrams, $Y_s$ for Edge BC is a sum of $Y_s$'s for Points B and C, for any $s \geq 0$. The following is a consequence of Eq. (15) and Corollary 1 in Appendix B.

$$\mathbb{E}\left[L(t)\right] \sim \frac{(1+\rho)p}{2}e^{-\frac{4H\sigma^2 t}{T}}, \qquad p \to \infty, \quad H/p \to \infty, \quad \rho = pH\sigma^4. \tag{23}$$

See Fig. 3, top left, for an experimental validation. In both cases, Eq. (22) and Eq. (23), the loss decays exponentially. This suggests that the ODE governing the weight evolution is, in fact, linear. For this reason, we call this regime *linearized training*.

In contrast to Point C, the above loss does not equal the loss of a zero model, $\frac{p}{2}$, at $t = 0$. This indicates that the model does not start to evolve from zero. This is the main qualitative difference between Point C and Edge BC.

### 6.3 Nonlinear underparameterized training: Edge EA

This model involves interactions with the target through diagrams $R$. As an underparameterized model, it only involves contracted diagrams having the form of "flowers" with "petals" consisting of a $p$-node and two edges of two possible colors. Merging with $R$ has the effect of recoloring one of the edges. For the diagram to admit a pairing without additional contractions, all petals must have same-color edges. Let $q$ denote the number of differently-colored petals, and consider a generalized generating function

$$f(x,y,z) = \sum_{s_R=0}^{\infty}\sum_{q=0}^{s_R}\sum_{s=\max(0,s_R-1)}^{\infty} C_{s_R,q,s}x^s y^{s_R} z^q, \tag{24}$$

where $C_{s_R,q,s}s!$ is the combinatorial factor corresponding to diagrams with $q$ differently-colored petals in expansion terms of $(\frac{1}{2}D - R)^{\star(s+1)}$ with exactly $s_R$ factors $R$. It is related to the loss function as follows.

$$\mathbb{E}[L(t)] \sim \frac{p}{2} + \rho^2 H f\left(-\frac{\rho t}{T}, \frac{1}{\rho}, 0\right), \qquad H \to \infty, \quad p/H \to \infty, \quad \rho = p\sigma^2. \tag{25}$$

The coefficients $C_{s_R,q,s}$ satisfy the following recurrence.

$$sC_{s_R,q,s} = -2(q+1)C_{s_R-1,q+1,s-1} - 2(s-s_R-q+3)C_{s_R-1,q-1,s-1} + 2(s-s_R+1)C_{s_R,q,s-1}. \tag{26}$$

Then Eq. (24) implies the following ODE for their generating function:

$$\Phi(\boldsymbol{x}) \cdot \nabla f(\boldsymbol{x}) = \phi(\boldsymbol{x})f(\boldsymbol{x}), \qquad \Phi(\boldsymbol{x}) = \begin{pmatrix} 1+2xyz-2x \\ 2(y-y^2z) \\ 2(y-yz^2) \end{pmatrix}, \qquad \phi(\boldsymbol{x}) = 4(1-yz), \tag{27}$$

where $\boldsymbol{x} = (x,y,z)$. The corresponding characteristic ODE is explicitly solvable; the characteristics are elliptic in the $yz$ plane (see Figure 4). Due to Eq. (25), we need to find $f$ at the plane $z = 0$. We know $f$ on the plane $x = 0$, since Eq. (24) implies $f(0,y,z) = \frac{1}{2} - yz$. We transfer the values of $f$ from the plane $x = 0$ to the plane $z = 0$ along the characteristics to obtain the explicit solution

$$f(x,y,0) = y^2\frac{1+2yr-r^2}{2(y-r)^2}, \qquad r = \sinh(2xy). \tag{28}$$

See Appendix C for a complete derivation. This together with Eq. (25) yields

$$\mathbb{E}[L(t)] \sim \frac{p}{2} + \frac{H}{2}\left(\frac{1+\rho^2}{\left(1 + \rho\sinh\left(\frac{2t}{T}\right)\right)^2} - 1\right), \qquad H \to \infty, \quad p/H \to \infty, \quad \rho = p\sigma^2. \quad (29)$$

See Fig. 3, top center, for an experimental validation. In contrast to the BC-limit solution Eq. (23), the loss does not decay exponentially, hence the training dynamics cannot be linear.

**Deriving Point A as a limit case of Edge EA.** We would expect to obtain the solution for Point A if we consider the limit of $\rho \to \infty$. Suppose $T \asymp 1$. Then for any finite $t > 0$, the above formula yields $\mathbb{E}[L(t)] \sim \frac{p}{2} + \frac{H}{2}\left(\operatorname{csch}^2(2t/T) - 1\right)$. That is, for large enough finite $t$, the loss already falls below $\frac{p}{2}$, therefore the target is learned, which should not happen at Point A.

At the same time, if $t = O_{\rho\to\infty}(1/\rho)$, we get $\mathbb{E}[L(t)] \sim \frac{p}{2} + \frac{H}{2}\frac{\rho^2}{(1+2\rho t/T)^2}$ as $\rho \to \infty$, which coincides with Eq. (20).

## 6.4 Nonlinear overparameterized training: Edge CD

We solve this model in a similar way as the underparameterized one above. It involves contracted diagrams which are again "flowers", but the petals consist of $H$ nodes instead of $p$ ones, while there is a $p$ node at the flower center. By performing similar steps as above, we arrive at the following asymptotic for the average loss:

$$\mathbb{E}[L(t)] \sim \frac{p}{2}\left(\frac{1+\rho^2}{1 + \rho^2\cosh\psi + \rho\sqrt{1+\rho^2}\sinh\psi}\right)^2, \qquad \psi = \frac{2t}{T}\sqrt{1+\rho^2}, \quad \rho = H\sigma^2, \quad (30)$$

as $p \to \infty$, $H/p \to \infty$. See Fig. 3, top right, for an experimental validation, and Appendix D for a complete derivation. One could easily check that Eq. (22) is a $\rho \to \infty$ limit of the above formula. Similarly to Edge EA, the loss does not decay exponentially, which indicates that the training dynamics is not linear in weights.

## 6.5 Vanishing initialization: Edge DE

Point E corresponds to $s_R = s+1$ for odd $s$, while $s_2 = s$ for even $s$, and $n = 1$. It is Pareto maximal when it dominates its neighbor points in the convex span, i.e. when for even $s$, $p^2H\sigma^4 \gg pH^2\sigma^4$ and $p^2H\sigma^4 \gg p^4H\sigma^8$, while for odd $s$, $pH\sigma^2 \gg pH^3\sigma^6$ and $pH\sigma^2 \gg p^3H\sigma^6$. This is equivalent to $p \gg H$, $p^2\sigma^4 \ll 1$, and $H^2\sigma^4 \ll 1$. Given first two, the last condition holds automatically, so we are left with $p \gg H$ and $p\sigma^2 \ll 1$. That is, we arrive into an underparameterized small initialization regime.

Similarly, Point D corresponds to $s_R = s + 1$ and $n = 1$ for odd $s$, while $s_R = s$ and $n = 2$ for even $s$. It is Pareto maximal when it dominates its neighbor points in the convex span, i.e. when for even $s$, $pH^2\sigma^4 \gg p^2H\sigma^4$ and $pH^2\sigma^4 \gg pH^4\sigma^8$, while for odd $s$, $pH\sigma^2 \gg pH^3\sigma^6$ and $pH\sigma^2 \gg p^3H\sigma^6$. This is equivalent to $H \gg p$, $H\sigma^2 \ll 1$, and $p\sigma^2 \ll 1$. Given first two, the last condition holds automatically, so we are left with $H \gg p$ and $H\sigma^2 \ll 1$. That is, we arrive into an overparameterized small initialization regime.

In order for points D and E to be of the same order, we need $p \asymp H$; let us take $H \sim \phi p$ for some finite $\phi > 0$. In order for these points to dominate their neighbors, we need $H\sigma^2 \ll 1$. That is, initialization has to be small, while dimensions have to grow proportionally. The following is a consequence of Eq. (15) and Corollary 2 in Appendix B.

$$\mathbb{E}\left[L(t)\right] \sim \frac{p}{2} - pH\sigma^2\sinh\left(\frac{2t}{T}\right) + \frac{3(1+\phi)}{4}p^2H\sigma^4\cosh\left(\frac{4t}{T}\right), \qquad p, H \to \infty, \quad \phi = H/p. \quad (31)$$

Though the sinh term dominates the cosh one for any finite $t/T$, for large enough $t/T$, the latter starts to prevail. One gets the solution for Point D by substituting $\phi = 0$, and that for Point E — by taking the limit $\phi = H/p \to \infty$.

We see the above solution does not make sense for large enough $t$. This does not disqualify the method we apply for the following reasons. First, the loss expansion wrt $t$ is valid only in the vicinity of $t = 0$. Second, interchanging the series and the expectation, as well as the series and the $p$ and $H$ limits, is not generally valid. We therefore expect the method to give reasonable results for small enough $t$.

See Fig. 3, bottom right, dotted lines, for an experimental validation. As we see, the analytical solution is indeed valid only up to a finite training time.

**Deriving Points D and E as limit cases of Edges CD and EA.** Similarly to the case of A as a limit case of EA, we would expect D and E to be vanishing $\rho$ limits of CD and EA, respectively. That is, the $\rho \to 0$ limit of Eq. (25) is given by

$$\mathbb{E}[L(t)] \sim \frac{p}{2} - Hp\sigma^2 \sinh\left(2t/T\right), \qquad H \to \infty, \quad p/H \to \infty, \quad p\sigma^2 \to 0. \tag{32}$$

This coincides with Eq. (31) when $p/H \to \infty$ and $p\sigma^2 \to 0$, for any finite $t/T$. Similarly, the $\rho \to 0$ limit of Eq. (30) is given by

$$\mathbb{E}[L(t)] \sim \frac{p}{2} \frac{1}{\left(1 + H\sigma^2 \sinh(2t/T)\right)^2}, \qquad p \to \infty, \quad H/p \to \infty, \quad H\sigma^2 \to 0. \tag{33}$$

This also coincides with Eq. (31) when $\phi = H/p \to \infty$ and $H\sigma^2 \to 0$, for any finite $t/T$. However, for large $t/T$, the above formula does not explode as Eq. (31), but converges to zero, as expected from an overparameterized model. We plot this curve as a blue dashed line in Fig. 3, bottom right. As we see, it matches the experiment much better than the corresponding dotted line.

## 7    DISCUSSION

**Summary.** The main contribution of the present paper is explicit solutions for the training dynamics of a linear autoencoder in the limit of large input and hidden sizes, and infinite training data. Our analysis is based on the methodology of Yarotsky et al. (2026). Although they provided a complete classification of limit regimes, they did not obtain any explicit solutions in our scenario. While the solution for BC corresponds to a well-known NTK limit (Jacot et al., 2018), which corresponds to linear training dynamics in weights, several other solutions we got (CD, DE, EA) correspond to nonlinear training dynamics; results of this type are very scarce in the literature.

**Future work.**    In the present work, we considered an infinite-data limit Eq. (3) of the general linear autoencoder loss Eq. (2). Nevertheless, the diagrammatic method of Yarotsky et al. (2026) could be applied directly to the finite-data loss function $L_X$. This case is particularly important as it allows us to study the phenomenon of *generalization* and *overfitting* by tracking both the evolution of $L$ and $L_X$ under the gradient flow of the latter.

The method of Yarotsky et al. (2026) could also be applied to the following cases. (1) deep autoencoder, (2) ridge-regularized loss, (3) gradient flow with momentum, (4) imbalanced initialization. We leave the classification of the learning regimes corresponding to the generalizations proposed above, akin to that provided by Theorem 1, as well as their explicit solutions, for future work.

We did not obtain exact solutions for free evolution in a balanced scenario, Edge AB, as it yields quite complicated combinatorics. We see this as a promising research direction on its own right.

ACKNOWLEDGMENTS

The research was supported by the Russian Science Foundation grant No. 25-11-00355

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

## A    LITERATURE OVERVIEW

### A.1    GRADIENT FLOW DYNAMICS FOR LINEAR NETWORKS

Linear networks, i.e. networks with identity activation functions, while being as expressive as simple linear models, exhibit non-trivial training dynamics. In particular, as noted in Saxe et al. (2013), for a small enough initialization magnitude, the loss decays in a ladder manner. It is conjectured that this happens due to the fact that gradient descent, starting from a saddle point at the origin, escapes it to be trapped in another saddle associated with the strongest eigencomponent of the data, then escapes it and gets trapped in another saddle. We observe a loss evolution which looks like a ladder, since gradient flow spends some time near each saddle before escaping it. This regime is therefore called *saddle-to-saddle*. We recover a regime similar to that discovered in Saxe et al. (2013) in our solution for Edge DE. Since the target is the identity matrix, the origin is the only saddle point the process encounters, and the "ladder" has only two steps, as can be seen in the corresponding plot of Fig. 3.

The solution of Saxe et al. (2013) relies on the assumption that the initial weights are aligned with the data. In contrast, Fukumizu (1998) obtains another solution relying on an assumption of initialization being *balanced*. Their solution was later revised by Braun et al. (2022) and generalized to imbalanced initialization by Dominé et al. (2024).

See also Tu et al. (2024) for an approximate solution for linear nets that interpolates between kernel and rich regimes.

### A.2    AUTOENCODERS

The idea of autoencoders (AE), i.e., neural networks trained to recover their own input, comes from the pioneering work of Rumelhart et al. (1985). Their applications include

1. Feature extraction. Since autoencoders are trained in an unsupervised manner, this can be used for semi-supervised learning (Erhan et al., 2010).

2. Anomaly detection (Vincent et al., 2008): an autoencoder fails to recover a sample that lies outside the distribution it was trained on; this can be used as a criterion for anomalies.

3. Denoising (Vincent et al., 2008): one could train an autoencoder not as an identity map but rather to remove noise from the input.

4. Dimensionality reduction (Hinton & Salakhutdinov, 2006): when the hidden size is smaller than the input/output dimension, a trained autoencoder keeps the most relevant components of the data.

An important special case is *linear autoencoders (LAE)* which we consider in the present work. LAEs were first introduced and approached theoretically in Baldi & Hornik (1989). Such models could be as well used for denoising (Pretorius et al., 2018) and for dimensionality reduction (Baldi & Hornik, 1989; Baldi, 2012; Kunin et al., 2019). The latter three works study the LAE loss landscape, while Pretorius et al. (2018) consider gradient flow dynamics under a special aligned weight initialization similar to that of Saxe et al. (2013). We emphasize that we do not exploit any kind of weight alignment in our analytic solutions.

Also related to our work, Cui & Zdeborová (2023) studied a general non-linear autoencoder in the limit of large and proportional input and dataset size, with hidden size bounded. They considered an exact solution but not the training dynamics.

### A.3    INFINITELY WIDE NETWORKS

As demonstrated in Jacot et al. (2018), the training process of a neural network becomes equivalent to a kernel method in the limit of infinite *width* under specific non-standard parameterization, where width is the minimal number of neurons over all hidden layers. The corresponding kernel is called *Neural Tangent Kernel (NTK)*, and was extensively studied in the literature. The NTK corresponds to Edge BC in our classification.

However, as observed empirically by Fort et al. (2020), the empirical NTK of realistic neural nets evolve during training, hence their training process is not equivalent to any kernel method. This is due to two reasons. First, realistic networks have finite size. Second, and probably more importantly, realistic networks are rarely parameterized in the way required for the NTK limit to hold.

This shows that infinite-width limits depend on parameterization. Golikov (2020b;a) provide a classification of such infinite-width limits for two-layered nets, while Yang & Hu (2021) provide an alternative classification for a much wider class of neural models.

Among various limits, there is another one, different from NTK, that attracted some attention in the literature. This limit is called *mean-field*, and was discovered independently in a number of works (Chizat & Bach, 2018; Rotskoff & Vanden-Eijnden, 2018; Sirignano & Spiliopoulos, 2020). Unfortunately, in contrast to NTK, the mean-field limit is quite difficult to work with. Moreover, it is naturally formulated for two-layered nets, but does not allow for a straightforward generalization to deeper nets. Nevertheless, there was a number of attempts to generalize it: Sirignano & Spiliopoulos (2022); Araújo et al. (2019); Nguyen (2019); Nguyen & Pham (2023). The mean-field limit corresponds to Edge CD in our classification.

## B  LEADING TERMS FOR SELECTED LIMIT REGIMES

### B.1  EDGE AB

**Proposition 1.** *Suppose $p, H \to \infty$ in such a way that $H/p \to 0$ and $p\sigma^2 \to 0$. Then $\forall s \geq 0$*

$$\mathbb{E}\left[\left(\frac{1}{2}D - R\right)^{\star(s+1)}\right] \sim 2^{s-1}(s+1)!p^{2+s}H\sigma^{4+2s}. \tag{34}$$

*Proof.* We would like to have as many p-type vertices after contraction. Any complete contraction of any diagram in $D^{\star(s+1)}$ is a tree with $2 + s$ double edges, hence with $3 + s$ vertices, we would equivalently like to minimize the number of H-type vertices. For any diagram in $D^{\star(s+1)}$, there is exactly one coupling that leads to a tree with exactly one H-type vertex. Indeed, the goal is to collapse all H-type vertices into one, while keeping all p-type nodes. To do this, we have to couple subsequent edges that share a p-type node. Such a procedure is always possible and leads to a single outcome no matter of the order of coupling.

Hence we simply need to compute the number of diagrams $N_s$ in $D^{\star(s+1)}$. We know that $N_0 = 1$. Then $N_s$ equals $N_{s-1}$ times the number of edges in each of the diagrams of $D^{\star s}$ times two: $N_s = 2(2 + 2s)N_{s-1} = 4(1 + s)N_{s-1}$. Resolving the recurrence gives the answer. □

**Proposition 2.** *Suppose $p, H \to \infty$ in such a way that $H/p \to \infty$ and $H\sigma^2 \to \infty$. Then $\forall s \geq 0$*

$$\mathbb{E}\left[\left(\frac{1}{2}D - R\right)^{\star(s+1)}\right] \sim 2^{2s-1}p^2H^{1+s}\sigma^{4+2s}. \tag{35}$$

*Proof.* Opposite to the proof of Proposition 1, we would like to keep as few p-type vertices after contraction as possible. There is no diagram in $D^{\star(s+1)}$ that could be coupled to a tree with only one p-type vertex. However, some of these diagrams could be coupled to a tree with two p-type vertices. Those are loops that could be divided into two disjoint paths of same type edges. Therefore we need to compute the number of loops $N_s$ of this type in $D^{\star(s+1)}$. We know that $N_0 = 1$. Then $N_s = 2 \times 4 \times N_{s-1} = 8N_{s-1}$. Resolving the recurrence gives the answer. □

### B.2  EDGE BC

**Proposition 3.** *Suppose $p, H \to \infty$ and $\sigma^2 \to 0$ in such a way that $H\sigma^2 \to \infty$, while $pH\sigma^4 \to 0$. Then*

$$\mathbb{E}\left[\left(\frac{1}{2}D - R\right)^{\star(s+1)}\right] = 2^{2s-1}\begin{cases} p^2H\sigma^4, & s = 0; \\ pH^s\sigma^{2s}, & s \geq 1. \end{cases} \tag{36}$$

*Proof.* We only consider terms $G_{i,j} = L_{a_0} \star L_{a_1} \star \ldots \star L_{a_s}$ for $0 \leq i < j \leq s$ with $a_i = a_j = 2$ and $a_k = 4 \; \forall k \notin \{i, j\}$. Our goal is to count the number of p-centreable loops in all such terms for a given $s$.

First consider the case of $s = 1$. Here the only maximal term is $G_{0,1} = R \star R$. It consists of two loops, both are p-centreable.

Now consider the case of $a_0 = a_1 = 2$, while $s \geq 2$. Since for any valid loop $G$, $R \star R \star G = 2 \times G$, this case does not result in any recoloring, hence gives zero p-centreable loops.

Suppose now either $a_0 = 4$, or $a_1 = 4$. By commutativity of $\star$, $a_0 = 4$ w.l.o.g. Suppose $a_i = a_j = 2$ for some $1 \leq i < j \leq s$. We claim that the only $j$ for which $G_{i,j}$ could have p-centreable loops is $j = s$.

Indeed, all loops in $L_{a_0} \star \ldots \star L_{a_{i-1}}$ have only H-type components. Next, all loops in $L_{a_0} \star \ldots \star L_{a_i}$ have H-type components and two odd components (necessarily next to each other). Then the same holds also for $L_{a_0} \star \ldots \star L_{a_{j-1}}$. Then all loops in $L_{a_0} \star \ldots \star L_{a_j}$ have at most two p-type components.

We want $G_{i,j}$ to have only p-type components. Since merging with $D$ does not introduce new p-type components, and does not destroy any components, we would like it to be able to introduce no new components at all (we do not want new H-type or odd ones). This is possible only when $L_{a_0} \star \ldots \star L_{a_j}$ has loops with trans-H-vertices. However, such loops necessarily have H-type or odd components. Therefore $L_{a_0} \star \ldots \star L_{a_j} \star D$ has no p-centreable loops. Hence $L_{a_j}$ has to be the last to merge, i.e. $j = s$.

Let us now compute the number of p-centreable loops in $G_{i,s}$ for $i \in [1 : s - 1]$; let us denote this number $N_i$. Since the maximal number of p-type components we could get is two, and merging with $D$ could not destroy any components, $\forall j \in [1 : s]$ we count only those loops in $L_{a_0} \star \ldots \star L_{a_j}$ that have only two components. Hence for $j < i$, we keep two trans-H-vertices, hence $L_{a_0} \star \ldots \star L_{a_{i-1}}$ has $8^{i-1}$ loops of interest. For each of them, merging with $L_{a_i}$ gives 4 loops with two (odd) components. In their turn, for each of them, merging with $L_{a_{i+1}}$ gives 4 loops that keep these two components and keep them odd; the same goes inductively for each $j \in [i + 1 : s - 1]$. The final merge with $L_{a_s}$ gives only two loops for each of them with those components transformed into p-type ones.

That is, $N_i = 8^{i-1} \times 4 \times 4^{s-i-1} \times 2 = 2^{2s+i-2} \; \forall i \in [1 : s - 1]$. By commutativity, $N_0 = N_1 = 2^{2s-1}$. Then the desired number of p-centreable loops $N$ in all $G_{i,s}$ for $i \in [0 : s - 1]$ is given by

$$N = \sum_{i=0}^{s} N_i = 2^{2s-1} + \sum_{i=1}^{s-1} 2^{2s+i-2} = 2^{2s-1} + 2^{2s-2+s} - 2^{2s-1} = 2^{3s-2}. \tag{37}$$

Since there are exactly $s - 1$ $D$-terms, we have to multiply by $2^{-s+1}$, which gives the answer $2^{2s-1}$. $\qquad \square$

The following is a corollary of Propositions 2 and 3.

**Corollary 1.** *Suppose $p, H \to \infty$ and $\sigma^2 \to 0$ in such a way that $H/p \to \infty$, while $pH\sigma^4 \to \rho$. Then*

$$\mathbb{E}\left[\left(\frac{1}{2}D - R\right)^{\star(s+1)}\right] = 2^{2s-1} \begin{cases} \rho p, & s = 0; \\ (1 + \rho)pH^s\sigma^{2s}, & s \geq 1. \end{cases} \tag{38}$$

*Proof.* Indeed, as apparent from the proofs of Proposition 2 and Proposition 3, the dominant diagrams for Points B and C do not intersect. Therefore, the corresponding $\mathbb{E}\left[\left(\frac{1}{2}D - R\right)^{\star(s+1)}\right]$ for Edge BC is nothing but a linear combination of that of Points B and C. $\qquad \square$

### B.3 EDGE DE

**Proposition 4.** *Suppose $p, H \to \infty$ and $\sigma^2 \to 0$ in such a way that $p/H \to \infty$, while $p\sigma^2 \to 0$. Then*

$$\mathbb{E}\left[\left(\frac{1}{2}D - R\right)^{\star(s+1)}\right] = pH\sigma^2 \begin{cases} \frac{p\sigma^2}{2}, & s = 0; \\ 3 \times 4^{s-1}p\sigma^2, & even \ s \geq 2; \\ 2^s, & odd \ s. \end{cases} \tag{39}$$

*Proof.* If $s$ is odd then $R^{\star(s+1)}$ consists of $2^s$ trivial loops.

If $s$ is even then we are interested in $R^{\star(s-k)} \star D \star R^{\star k}$ for all $k \in [0:s]$. If $k = 0$ then this equals $2^{s-1} \times 4 \times D = 2^{s+1} \times D$, which could be contracted to $2^{s+1}$ trees with one h-type vertex.

Suppose $k$ is even in $[2:s-2]$. Then $R^{\star(s-k)} \star D = 2^{s-k-1} \times 4 \times D = 2^{s-k+1} \times D$. At the same time, among all $4^k$ loops in $D \star R^{\star k}$, only a half could be contracted to a tree with one h-type vertex. Then $R^{\star(s-k)} \star D \star R^{\star k}$ gives $2^{s-k+1} \times 2^{2k-1} = 2^{s+k}$ such loops.

If $k = s$ then we get $2^{2s-1}$ such loops.

Suppose $k$ is odd. Then $R^{\star(s-k)} \star D = 2^{s-k-1} \times R \star D = 2^{s-k-1} \times D \star R$. Then $D \star R \star R^{\star k} = D \star R^{\star(k+1)}$, and $k+1$ is even. Therefore $R^{\star(s-k)} \star D \star R^{\star k}$ gives $2^{s-k-1} \times 2^{2k+1} = 2^{s+k}$ "good" loops, the same number as for even $k$.

Summing over all $k \in [1:s-1]$ gives $2^s \times (2^s - 2) = 2^{2s} - 2^{s+1}$. Adding the cases of $k = 0$ and $k = s$ results in $2^{2s} - 2^{s+1} + 2^{2s-1} + 2^{s+1} = 3 \times 2^{2s-1}$ "good" loops for even $s$. Because of $1/2$ factor at $D$, these terms contribute $\frac{3}{4}2^{2s}$. $\qquad\square$

**Proposition 5.** *Suppose $p, H \to \infty$ and $\sigma^2 \to 0$ in such a way that $H/p \to \infty$, while $H\sigma^2 \to 0$. Then*

$$\mathbb{E}\left[\left(\frac{1}{2}D - R\right)^{\star(s+1)}\right] = pH\sigma^2 \begin{cases} \frac{p\sigma^2}{2}, & s = 0; \\ \left(3 \times 4^{s-1} - 2^s\right)H\sigma^2, & even \ s \geq 2; \\ 2^s, & odd \ s. \end{cases} \tag{40}$$

*Proof.* The proof is similar to that of Proposition 4, but the case of $k = 0$ and even $s$ gives no trees with one i/o-type vertex. Therefore the result is smaller by $2^s$ for even $s$. $\qquad\square$

**Corollary 2.** *Suppose $p, H \to \infty$ and $\sigma^2 \to 0$ in such a way that $H/p \to \phi$, while $H\sigma^2 \to 0$. Then*

$$\mathbb{E}\left[\left(\frac{1}{2}D - R\right)^{\star(s+1)}\right] = pH\sigma^2 \begin{cases} \frac{p\sigma^2}{2}, & s = 0; \\ \left(3 \times 4^{s-1}(1+\phi) - 2^s\phi\right)p\sigma^2, & even \ s \geq 2; \\ 2^s, & odd \ s. \end{cases} \tag{41}$$

*Proof.* In contrast to Edge BC, the optimal diagrams for Points D and E intersect in some cases. Precisely, as apparent from the proofs of Proposition 5 and Proposition 4, the optimal diagrams for $s = 0$ and odd $s$ are exactly the same for both cases. On the other hand, the optimal diagrams do not intersect for positive even $s$. In this case, the corresponding $\mathbb{E}\left[\left(\frac{1}{2}D - R\right)^{\star(s+1)}\right]$ for Edge DE is a linear combination of that of Points D and E. $\qquad\square$

## C LOSS EVOLUTION FOR EDGE EA

**The diagram expansion.** Recall our general diagram expansion Eq. (15):

$$\mathbb{E}[L(t)] \sim \frac{p}{2} + \sum_{s=0}^{\infty} \mathbb{E}\left[\left(\frac{1}{2}D - R\right)^{\star(s+1)}\right] \frac{(-t)^s}{T^s s!}. \tag{42}$$

The edges in the diagrams $D$ and $R$ can be of two colors, $m = 1, 2$. Merging a given diagram $G$ with diagram $D$ over a particular edge $u_{ki}^{(m)}$ in $G$, where $k$ is a $H$-node, $i$ is a $p$-node and $m$ is a

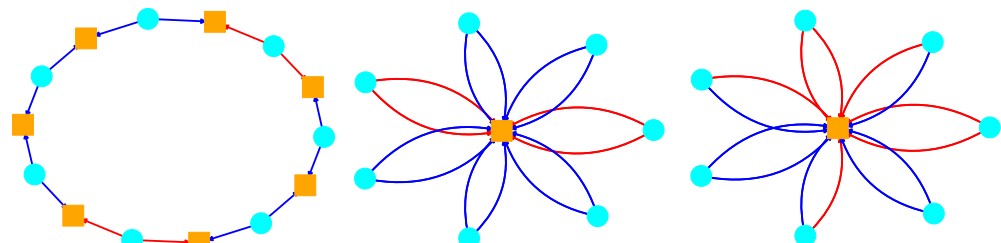

Figure 5: Left to right: (1) An uncontracted diagram from the free evolution $D^{\star s}$. It has a circular form with alternating $s+1$ $p$-nodes and $s+1$ $H$-nodes. Each $p$-node has same-color edges. The provided example is for $s = 6$. (2) An optimally contracted diagram for the free underparameterized regime: a "flower" with one $H$-node and $s+1$ "petals". (3) Contracted non-free underparameterized diagrams from $(\frac{1}{2}D - R)^{\star s}$. Due to merging with $R$, some edges are recolored. When there are petals with differently-colored edges (as in this example), the diagram does not admit an edge pairing without extra contractions.

color, replaces this edge by a sequence of three edges, thereby adding two new nodes $k', i'$ and two edges of another color:

$$u_{ki}^{(m)} \rightsquigarrow u_{ki'}^{(3-m)} u_{k'i'}^{(3-m)} u_{k'i}^{(m)}. \tag{43}$$

Thus, diagrams obtained by repeated mergers of diagrams $D$ (i.e., "free" diagrams) have a circular shape with alternating $H$- and $p$-nodes, with the edges attached to $p$-nodes forming same-color pairs (Fig. 5 (c)).

Merging a given diagram $G$ with diagram $R$ over a particular edge in $G$ recolors this edge. As a result, "nonfree' diagrams of $(\frac{1}{2}D - R)^{\star(s+1)}$ still have a circular shape, but without the color constraint. Considering the binomial expansion of $(\frac{1}{2}D - R)^{\star(s+1)}$ into variously-ordered sequences of mergers of $s_D$ diagrams $D$ and $s_R$ diagrams $R$ with $s_D + s_R = s + 1$, the resulting circles have length $2(s_D + 1)$.

**Diagrams in the underparameterized regime.** By Theorem 1 and discussion in Section 5, the underparameterized asymmetric model is represented by Pareto-optimal terms (19) with $n = 1$:

$$p^{s_D+1} H \sigma^{2(s_D+1)}. \tag{44}$$

These terms result from contractions of diagrams in $(\frac{1}{2}D - R)^{\star(s+1)}$ into "flowers" with a single $H$-node (i.e., the $H$-nodes are fully contracted, while the $p$-nodes remain uncontracted), see Fig. 5. The flower has $s_D + 1$ "petals", each consisting of a $p$-node connected to the contracted $H$-node by two edges, possibly of two diferent colors. If there is at least one petal with differently-colored edges, the diagram does not admit an edge pairing (without extra contractions) and so does not contribute to the leading term.

Thus, we are interested only in those contracted flower diagrams in $(\frac{1}{2}D - R)^{\star(s+1)}$ in which petals do not have differently-colored edges. However, the combinatorial coefficients associated with such special diagrams are not simply related to each other. It is convenient to consider diagrams with $q$ petals having differently-colored edges, for general $q$. This more general point of view allows to simplify the connection between combinatorial coefficients, leading to a first-order PDE for the generating function.

**The generalized generating function and its connection to the loss.** Motivated by above considerations, we introduce the generating function

$$f(x, y, z) = \sum_{s_R=0}^{\infty} \sum_{q=0}^{s_R} \sum_{s=\max(0,s_R-1)}^{\infty} C_{s_R,q,s} x^s y^{s_R} z^q. \tag{45}$$

Here, we define $C_{s_R,q,s}$ so that $C_{s_R,q,s}s!$ is the total numerical coefficient corresponding to those terms in the binomial expansion of $(\frac{1}{2}D - R)^{\star(s+1)}$ that include $s_R$ factors $R$ and result in flower diagrams with exactly $q$ differently-colored petals. The coefficient $C_{s_R,q,s}$ absorbs the factors $\frac{1}{2}$

and $-1$ appearing in $\frac{1}{2}D - R$. Summation in $f$ is restricted to $0 \leq q \leq s_R$ because, clearly, $q$ differently-colored petals require at least $q$ recoloring of edges, i.e. $s_R \geq q$.

In our approximation of keeping only the Pareto-optimal underparameterized terms (44), the loss (42) can be written in terms of the g.f. $f$ by noting that taking only the $q = 0$ terms in $f$ corresponds to setting $z = 0$:

$$\mathbb{E}[L(t)] \sim \frac{p}{2} + \sum_{s=0}^{\infty} \sum_{s_D=0}^{s+1} C_{s+1-s_D, q=0, s} p^{s_D+1} H\sigma^{2(s_D+1)} \frac{(-t)^s}{T^s s!} \tag{46}$$

$$= \frac{p}{2} + \sum_{s=0}^{\infty} \sum_{s_R=0}^{s+1} C_{s_R, q=0, s} p^{s+2-s_R} H\sigma^{2(s+2-s_R)} \frac{(-t)^s}{T^s s!} \tag{47}$$

$$= \frac{p}{2} + p^2 H\sigma^4 f(-p\sigma^2 t/T, 1/(p\sigma^2), 0). \tag{48}$$

**The recurrence.** Diagrams in $(\frac{1}{2}D - R)^{\star(s+1)}$ are obtained from diagrams in $(\frac{1}{2}D - R)^{\star s}$ by merging with $D$ or $R$. In the former case, the number $q$ of petals with differently-colored edges does not change, while in the latter case it is either increased by one (if we merge over a same-colored petal) or decreased by one (if we merge over a differently-colored petal). By counting the relevant edges and taking into account the necessary coefficients, we get the recurrence

$$sC_{s_R, q, s} = -2(q+1)C_{s_R-1, q+1, s-1} - 2(s-s_R-q+3)C_{s_R-1, q-1, s-1} + 2(s-s_R+1)C_{s_R, q, s-1}. \tag{49}$$

Importantly, this recurrence holds for all $s_R, q, s \in \mathbb{Z}$. At the initial value $s = 0$ the reduction to $s - 1$ described above is not applicable, but the recurrence still holds since both the l.h.s. and r.h.s. vanish. The values $C_{s_R, q, s}$ do not vanish only if $0 \leq q \leq s_R \leq s + 1, s \geq 0$, and $q$ and $s_R$ have the same parity.

**The PDE and reduction to ODE.** The recurrence yields the differential equation

$$x\partial_x f = 2x[-yz^{-1}z\partial_z - yz(x\partial_x - y\partial_y - z\partial_z + 2) + (x\partial_x - y\partial_y + 2)]f, \tag{50}$$

or equivalently

$$[(1 + 2xyz - 2x)\partial_x + 2(y - y^2 z)\partial_y + 2(y - yz^2)\partial_z]f = 4(1 - yz)f. \tag{51}$$

This is a 1'st order PDE that can be solved by the method of characteristics. Denote $\mathbf{x} = (x, y, z)$ and write the PDE as

$$\Phi(\mathbf{x})^T \nabla_F(\mathbf{x}) = \phi f(\mathbf{x}) \tag{52}$$

with the vector field

$$\Phi(\mathbf{x}) = \begin{pmatrix} 1 + 2xyz - 2x \\ 2(y - y^2 z) \\ 2(y - yz^2) \end{pmatrix} \tag{53}$$

and function

$$\phi(\mathbf{x}) = 4(1 - yz). \tag{54}$$

Then the solution at a given point $\mathbf{x}_0$ can be found as

$$f(\mathbf{x}_0) = f(\mathbf{x}_1) e^{-\int_{\tau_0}^{\tau_1} \phi(\mathbf{x}(\tau)) d\tau}, \tag{55}$$

where $\mathbf{x}(\tau)$ is an integral curve of the field $\Phi$ connecting the point $\mathbf{x}_0$ to the point $\mathbf{x}_1$:

$$\dot{\mathbf{x}} = \Phi(\mathbf{x}), \quad \mathbf{x}(\tau_0) = \mathbf{x}_0, \quad \mathbf{x}(\tau_1) = \mathbf{x}_1. \tag{56}$$

By Eq. (48), finding the loss requires us to find the values of $f$ on the hyperplane $z = 0$. On the other hand, the only terms present in $f$ with $s = 0$ are $C_{s_R=0, q=0, s=0} = \frac{1}{2}$ and $C_{s_R=1, q=1, s=0} = -1$, so $f$ has a simple form on the hyperplane $x = 0$:

$$f(0, y, z) = C_{0,0,0} + C_{1,1,0} yz = \frac{1}{2} - yz. \tag{57}$$

Accordingly, we will obtain the solution $f$ on the hyperlane $z = 0$ by transfering $f$ from the plane $x = 0$ along characteristics.

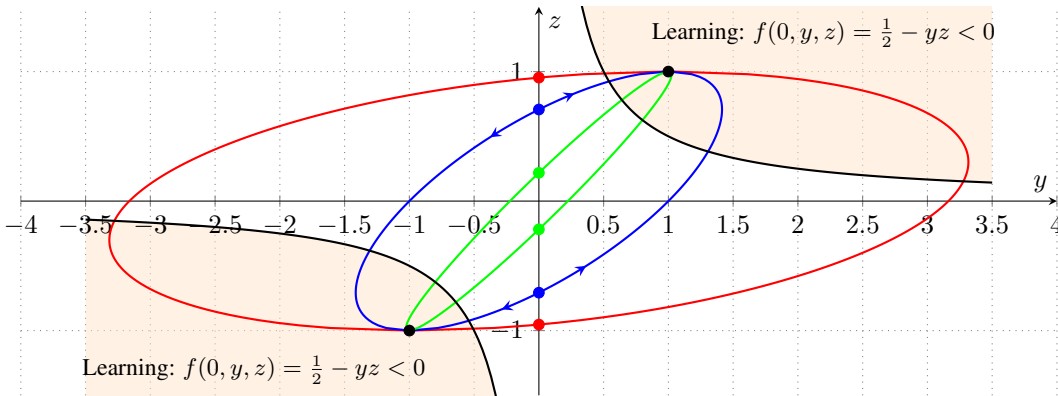

Figure 6: ODE (52) in the $yz$ plane.

**Analysis of the ODE.** The ODE (52) has subsystem $\{y, z\}$ independent of $x$. The evolution in this subsystem has the first integral

$$I = \frac{1 - z^2}{(y - z)^2}, \tag{58}$$

showing that the phase curves in the $yz$ plane are arcs of ellipses touching the lines $z = \pm 1$ at the points $(y, z) = \pm(1, 1)$ (see Figure 6).

Another first integral, involving all three variables, is

$$J = 4xy + \ln \frac{1 - z}{1 + z}. \tag{59}$$

We can also note that

$$\frac{d}{d\tau} \ln y = 2(1 - yz), \tag{60}$$

so that

$$\int \phi(\mathbf{x}(\tau)) d\tau = 4 \int (1 - yz) d\tau = 2 \ln y + c. \tag{61}$$

It follows that we can write the multiplier $e^{-\int_{\tau_0}^{\tau_1} \phi(\mathbf{x}(\tau)) d\tau}$ appearing in the solution (55) simply as

$$e^{-\int_{\tau_0}^{\tau_1} \phi(\mathbf{x}(\tau)) d\tau} = \frac{y_0^2}{y_1^2}. \tag{62}$$

**The solution.** By Eqs. (56) and (61), we have

$$f(x, y, 0) = \left(\frac{1}{2} - y_1 z_1\right) \frac{y^2}{y_1^2} \tag{63}$$

along an integral curve connecting the points

$$\mathbf{x}_0 = (x, y, 0), \quad \mathbf{x}_1 = (0, y_1, z_1). \tag{64}$$

It remains to express $y_1, z_1$ in terms of $x, y$. From the first integral $J$ we get

$$z_1 = \frac{e^{-4xy} - 1}{e^{-4xy} + 1}. \tag{65}$$

Then from the first integral $I$ we get

$$y_1 = y\sqrt{1 - z_1^2} + z_1 = y\frac{2e^{-2xy}}{1 + e^{-4xy}} - \frac{1 - e^{-4xy}}{1 + e^{-4xy}} = \frac{e^{-4xy} + 2ye^{-2xy} - 1}{1 + e^{-4xy}}. \tag{66}$$

It follows that

$$f(x, y, 0) = \left(\frac{1}{2} - y_1 z_1\right) \frac{y^2}{y_1^2(\tau_{x,y})} \tag{67}$$

$$= \frac{y^2[(1 + e^{-4xy})^2 + 2(1 - e^{-4xy})(e^{-4xy} + 2ye^{-2xy} - 1)]}{2(e^{-4xy} + 2ye^{-2xy} - 1)^2} \tag{68}$$

$$= y^2 \frac{1 + 2yr - r^2}{2(y - r)^2}, \qquad r = \sinh(2xy). \tag{69}$$

**Interpretation of learning.** The above analysis and Figure 6 provide a simple interpretation of learning in this model. We are interested in the values $f(x, y, 0)$ that correspond to the $y$ axis in Figure 6. Different values of $x$ correspond to different points along the elliptic arc going through the point $(y, 0)$. In the context of loss function (48), the values $y$ are positive, since $y = 1/(p\sigma^2)$ and the sign of $y$ is preserved by the dynamics. On the other hand, the initial $x = -p\sigma^2 t/T < 0$. Taking larger time $t$ corresponds to going along the arc towards the point $(y, z) = (1, 1)$. At sufficiently large $t$ (i.e., large negative $x$) we get into the region $\frac{1}{2} - yz < 0$. Then, by (55) or (63), $f(x, y, 0)$ becomes negative, meaning that the loss becomes less than the loss $\frac{p}{2}$ of the trivial zero model. That makes it natural to identify the region $\frac{1}{2} - yz < 0$ as the *learning region*.

As $t \to +\infty$ (i.e., $x \to -\infty$), the characteristic trajectories converge to $(y, z) = (1, 1)$ and

$$\lim_{x \to -\infty} f(x, y, 0) = -\frac{y^2}{2}. \tag{70}$$

It follows by (48) that

$$\lim_{t \to +\infty} \mathbb{E}[L(t)] \sim \frac{p}{2} - \frac{H}{2}, \tag{71}$$

which is the natural and expected result: using a model of tensor rank $H$ allows us to learn $H$ out of $p$ nonzero components of the target tensor $I$.

## D  LOSS EVOLUTION FOR EDGE CD

In order to compute the loss, we need to count the total number of trees with exactly one p-type vertex that result from contraction of loops in $\left(\frac{1}{2}D - R\right)^{\star(s+1)}$ for all $s \geq 0$. These trees are "flowers" with a $p$-vertex at the center and $H$-vertices at the "petals". We call loops that could be contracted to such flowers *p-contractible*. These flowers look like the flowers in Figure 4 for the underparametrized regime, but with exchanged roles of $p$- and $H$-nodes.

We call a vertex adjacent to edges of different colors *trans vertex*. A loops is $p$-contractible iff it has no trans-$H$ vertices.

**Generating function.** Let us introduce the following generating function:

$$f_{s_R, q}(x) = \sum_{s = \max(0, s_R - 1)}^{\infty} C_{s_R, q, s} x^s, \tag{72}$$

where $C_{s_R, q, s}$ is defined as follows. Let

$$\frac{1}{s!} \left(\frac{1}{2}D - R\right)^{\star(s+1)} = \sum_{s_R = 0}^{s+1} \sum_{G \in \mathcal{G}_{s_R, s}} c_G G, \tag{73}$$

where $\mathcal{G}_{s_R, s}$ is a set of all loops in the above expression resulted from using exactly $s_R$ $R_2$ terms. Then we define

$$C_{s_R, q, s} = \sum_{\substack{G \in \mathcal{G}_{s_R, s}: \\ G \text{ has } q \text{ trans-}H \text{ vertices}}} c_G. \tag{74}$$

When $H\sigma^2 \sim \rho$, while $H/p \to \infty$, we are interested only in $p$-contractible loops. Since a loop is $p$-contractible iff it has no trans-$H$ vertices, the expected loss is given by

$$\mathbb{E}[L(t)] \sim \frac{p}{2} + pH^2\sigma^4 \sum_{s_R=0}^{\infty} \frac{f_{s_R,0}(-H\sigma^2 t/T)}{H^{s_R}\sigma^{2s_R}} \sim \frac{p}{2} + p\rho^2 \sum_{s_R=0}^{\infty} \frac{f_{s_R,0}(-\rho t/T)}{\rho^{s_R}}, \tag{75}$$

where $\rho = H\sigma^2$.

**Recursive relation for C.**   We have the following recurrence:

$$
\begin{aligned}
sC_{s_R,q,s} = &-2(q+1)C_{s_R-1,q+1,s-1} - 2(s+2-s_R-(q-1))C_{s_R-1,q-1,s-1} \\
&+ 2qC_{s_R,q,s-1} + 2(s+1-s_R-(q-2))C_{s_R,q-2,s-1} \\
= &-2(q+1)C_{s_R-1,q+1,s-1} - 2(s-s_R-q+3)C_{s_R-1,q-1,s-1} \\
&+ 2qC_{s_R,q,s-1} + 2(s-s_R-q+3)C_{s_R,q-2,s-1}.
\end{aligned}
\tag{76}
$$

By construction, $C_{s_R,q,s} = 0$ whenever $s < 0$, or $s_R > s+1$, or $s_R < 0$, or $q < 0$.

**Generalized generating function.**   Let us intoduce the following generalized generating function of three variables:

$$f(x,y,z) = \sum_{q=0}^{\infty}\sum_{s_R=0}^{\infty} f_{s_R,q}(x)y^{s_R}z^q = \sum_{q=0}^{\infty}\sum_{s_R=0}^{\infty}\sum_{s=\max(0,s_R-1)}^{\infty} C_{s_R,q,s}x^s y^{s_R} z^q. \tag{77}$$

Given the above, the loss is expressed as follows:

$$\mathbb{E}[L(t)] \sim \frac{p}{2} + p\rho^2 \sum_{s_R=0}^{\infty} \frac{f_{s_R,0}(-\rho t/T)}{\rho^{s_R}} = \frac{p}{2} + p\rho^2 f\left(-\frac{\rho t}{T}, \frac{1}{\rho}, 0\right). \tag{78}$$

The main recurrence Eq. (76) yields the following ODE:

$$
\begin{aligned}
x\partial_x f &+ 2xy\partial_z f \\
&+ 2xyz\left(x\partial_x - y\partial_y - z\partial_z + 2\right)f - 2xz\partial_z f - 2xz^2\left(x\partial_x - y\partial_y - z\partial_z + 2\right)f = 0,
\end{aligned}
\tag{79}
$$

or, equivalently,

$$
\begin{aligned}
\left(1 + 2xyz - 2xz^2\right)\partial_x f &+ \left(-2y^2 z + 2yz^2\right)\partial_y f \\
&+ \left(2y - 2yz^2 - 2z + 2z^3\right)\partial_z f = \left(-4yz + 4z^2\right)f.
\end{aligned}
\tag{80}
$$

This is a first-order ODE which could be expressed as follows:

$$\Phi(\boldsymbol{x})\cdot\nabla f(\boldsymbol{x}) = \phi(\boldsymbol{x})f(\boldsymbol{x}), \tag{81}$$

where $\boldsymbol{x} = (x,y,z)$, and

$$\Phi(\boldsymbol{x}) = \begin{pmatrix} 1 + 2xz(y-z) \\ -2yz(y-z) \\ 2(1-z^2)(y-z) \end{pmatrix}, \qquad \phi(\boldsymbol{x}) = -4z(y-z). \tag{82}$$

Then the solution at a given point $\boldsymbol{x}_0$ is given by

$$f(\boldsymbol{x}_0) = f(\boldsymbol{x}_1)e^{-\int_{\tau_0}^{\tau_1}\phi(\boldsymbol{x}(\tau))\,d\tau}, \tag{83}$$

where the integral curve $\boldsymbol{x}(\tau)$ is defined by

$$\dot{\boldsymbol{x}}(\tau) = \Phi(\boldsymbol{x}(\tau)), \qquad \boldsymbol{x}(\tau_1) = \boldsymbol{x}_1, \quad \boldsymbol{x}(\tau_0) = \boldsymbol{x}_0. \tag{84}$$

We are interested in the solution at $z = 0$. We already know the solution at $x = 0$:

$$f(0,y,z) = C_{0,2,0}z^2 + C_{1,1,0}yz = \frac{z^2}{2} - yz = \frac{z}{2}(z - 2y). \tag{85}$$

**The $\{y, z\}$ subsystem.** The $\{y, z\}$ subsystem is independent on $x$ and allows for the following first integral:

$$I = \frac{y^2}{1 - z^2}. \tag{86}$$

The phase curves are ellipses for $I > 0$, or hyperbolae for $I < 0$, $y^2 + Iz^2 = I$ with common points $(y, z) = (0, \pm 1)$. Since we interested in curves passing $z = 0$, $I$ has to be non-negative. If we take $z_0 = 0$ then $I = y_0^2$.

For the loss to decrease, we need $f(x, y, 0)$ to be negative. For this, we need $f(\boldsymbol{x}(\tau))$ to stay negative along the integral curve. Suppose $x(\tau) = 0$ for some $\tau^* \in (\tau_0, \tau_1)$. Then $f(0, y(\tau^*), z(\tau^*))$ has to be negative. Then $z(\tau^*) < 2y(\tau^*)$ whenever $z(\tau^*) > 0$, and $z(\tau^*) > 2y(\tau^*)$ otherwise. In particular, $z(\tau^*)$ and $y(\tau^*)$ have to be of the same sign. Since $z = y$ is a stationary point, $z(\tau)$ and $y(\tau)$ keep their (same) signs all along the way. Therefore we could rewrite the above first integral as

$$y = y_0 \sqrt{1 - z^2}. \tag{87}$$

Note that

$$\phi(\boldsymbol{x}) = -4z(y - z) = 2\frac{\dot{y}}{y} = 2\frac{d \ln y}{d\tau}. \tag{88}$$

Therefore

$$e^{-\int_{\tau_0}^{\tau_1} \phi(\boldsymbol{x}(\tau)) \, d\tau} = \frac{y_0^2}{y_1^2} = \frac{1}{1 - z_1^2}. \tag{89}$$

**General $(y, z)$-solution.**

$$\dot{y} = -2yz(y - z), \qquad \dot{z} = 2(1 - z^2)(y - z). \tag{90}$$

Given the above first integral,

$$\dot{z} = 2(1 - z^2)\left(y_0\sqrt{1 - z^2} - z\right). \tag{91}$$

This gives the solution in implicit form:

$$\tau_1 - \tau_0 = \int_{z_0}^{z_1} \frac{dz}{2(1 - z^2)\left(y_0\sqrt{1 - z^2} - z\right)} = \frac{1}{4} \ln\left(\frac{1 - z_1^2}{1 - z_0^2}\right) - \frac{1}{2} \ln\left(\frac{y_0\sqrt{1 - z_1^2} - z_1}{y_0\sqrt{1 - z_0^2} - z_0}\right). \tag{92}$$

We are ultimately interested in the case of $z_0 = 0$:

$$\tau_1 - \tau_0 = \frac{1}{4} \ln\left(1 - z_1^2\right) - \frac{1}{2} \ln\left(\sqrt{1 - z_1^2} - z_1/y_0\right) = \frac{1}{4} \ln\left(\frac{1 - z_1^2}{\left(\sqrt{1 - z_1^2} - z_1/y_0\right)^2}\right). \tag{93}$$

This gives

$$e^{4(\tau_1 - \tau_0)} = \frac{1}{(1 - u_1)^2}, \qquad u_1 = \frac{z_1}{y_0\sqrt{1 - z_1^2}}. \tag{94}$$

Hence $u_1 = 1 \pm e^{-2(\tau_1 - \tau_0)}$. We know that if $\tau_1 = \tau_0$ then $z_1 = z_0 = 0$, therefore $u_1 = 0$. Then the desired root has a minus sign:

$$u_1 = 1 - e^{-2(\tau_1 - \tau_0)}. \tag{95}$$

This gives a linear equation for $z_1^2$:

$$\frac{z_1^2}{y_0^2(1 - z_1^2)} = \left(1 - e^{-2(\tau_1 - \tau_0)}\right)^2; \tag{96}$$

$$z_1^2 = \frac{a^2}{1 + a^2}, \quad 1 - z_1^2 = \frac{1}{1 + a^2}, \qquad a = y_0\left(1 - e^{-2(\tau_1 - \tau_0)}\right). \tag{97}$$

Therefore assuming $\tau_0 = 0$, the general solution for $z(0) = 0$, $y(0) = y_0$ is given by

$$z^2(\tau) = \frac{y_0^2\left(1 - e^{-2\tau}\right)^2}{1 + y_0^2\left(1 - e^{-2\tau}\right)^2}, \qquad y^2(\tau) = \frac{y_0^2}{1 + y_0^2\left(1 - e^{-2\tau}\right)^2}. \tag{98}$$

**The $x$ dynamics.**   Note that

$$\frac{d(xy)}{d\tau} = \dot{x}y + x\dot{y} = y. \tag{99}$$

Therefore

$$x(\tau_1)y(\tau_1) - x_0 y_0 = \int_0^{\tau_1} y(\tau)\, d\tau. \tag{100}$$

**The final solution.**   Since we already know the solution for $x = 0$, we take $x(\tau_1) = 0$. Then the above formula gives a condition for $\tau_1$:

$$-x_0 y_0 = \int_0^{\tau_1} y(\tau)\, d\tau = y_0 \int_0^{\tau_1} \frac{d\tau}{\sqrt{1 + y_0^2\left(1 - e^{-2\tau}\right)^2}}$$

$$= \frac{y_0}{2\sqrt{1 + y_0^2}} \left[ \tanh^{-1}\left( \frac{1 + y_0^2\left(1 - e^{-2\tau_1}\right)}{\sqrt{1 + y_0^2}\sqrt{1 + y_0^2\left(1 - e^{-2\tau_1}\right)^2}} \right) - \tanh^{-1}\left( \frac{1}{\sqrt{1 + y_0^2}} \right) \right]; \tag{101}$$

$$\tanh\left( -2x_0\sqrt{1 + y_0^2} + \tanh^{-1}\left( \frac{1}{\sqrt{1 + y_0^2}} \right) \right) = \frac{1 + y_0^2\left(1 - e^{-2\tau_1}\right)}{\sqrt{1 + y_0^2}\sqrt{1 + y_0^2\left(1 - e^{-2\tau_1}\right)^2}}. \tag{102}$$

Introducing $u = y_0\left(1 - e^{-2\tau_1}\right)$, we get a quadratic equation on $u$:

$$a^2 = \frac{(1 + y_0 u)^2}{1 + u^2}, \qquad a = \sqrt{1 + y_0^2}\tanh\left( -2x_0\sqrt{1 + y_0^2} + \tanh^{-1}\left( \frac{1}{\sqrt{1 + y_0^2}} \right) \right); \tag{103}$$

$$(y_0^2 - a^2)u^2 + 2y_0 u + (1 - a^2) = 0; \tag{104}$$

$$u = \frac{-y_0 \pm \sqrt{y_0^2 - (y_0^2 - a^2)(1 - a^2)}}{y_0^2 - a^2} = \frac{-y_0 \pm \sqrt{a^2 - a^4 + y_0^2 a^2}}{y_0^2 - a^2} = \frac{-y_0 \pm a\sqrt{1 - a^2 + y_0^2}}{y_0^2 - a^2}. \tag{105}$$

If we take $x_0 = 0$ then $a = 1$, while $\tau_1$ has to be zero, which means that the expected solution is $u = 0$. Then we have to choose the "plus" sign. Plugging back $a$,

$$u = \frac{-y_0 + a\sqrt{1 + y_0^2}\,\mathrm{sech}\,(v_0)}{(1 + y_0^2)\,\mathrm{sech}^2\,(v_0) - 1}$$

$$= \frac{-y_0\cosh^2(v_0) + (1 + y_0^2)\sinh(v_0)}{1 + y_0^2 - \cosh^2(v_0)} = \frac{-y_0\cosh^2(v_0) + (1 + y_0^2)\sinh(v_0)}{y_0^2 - \sinh^2(v_0)}, \tag{106}$$

$$v_0 = -2x_0\sqrt{1 + y_0^2} + \tanh^{-1}\left( \frac{1}{\sqrt{1 + y_0^2}} \right). \tag{107}$$

Then

$$f(\boldsymbol{x}(\tau_0)) = f(\boldsymbol{x}(\tau_1))e^{-\int_{\tau_0}^{\tau_1}\phi(\boldsymbol{x}(\tau))\,d\tau} = \frac{y_0^2}{y^2(\tau_1)}\left( \frac{z^2(\tau_1)}{2} - y(\tau_1)z(\tau_1) \right)$$

$$= y_0^2\left( \frac{z^2(\tau_1)}{2y^2(\tau_1)} - \frac{z(\tau_1)}{y(\tau_1)} \right) = y_0^2\left( \frac{1}{2}\left(1 - e^{-2\tau_1}\right)^2 - \left(1 - e^{-2\tau_1}\right) \right) = \frac{u^2}{2} - y_0 u. \tag{108}$$

**Explicit loss expression.**

$$\mathbb{E}[L(t)] \sim \frac{p}{2} + p\rho^2 f\left( -\frac{\rho t}{T}, \frac{1}{\rho}, 0 \right) = \frac{p}{2}\left(1 + \rho^2 u^2 - 2\rho u\right) = \frac{p}{2}(1 - \rho u)^2, \tag{109}$$

where

$$u = \frac{-\rho\cosh^2 v + (1 + \rho^2)\sinh v}{1 - \rho^2\sinh^2 v}, \qquad v = \frac{2t}{T}\sqrt{1 + \rho^2} + \tanh^{-1}\left( \frac{\rho}{\sqrt{1 + \rho^2}} \right). \tag{110}$$

This could be further simplified:

$$
\begin{aligned}
1 - \rho u &= \frac{1 - \rho^2 \sinh^2 v + \rho^2 \cosh^2 v - \rho(1+\rho^2)\sinh v}{1 - \rho^2 \sinh^2 v} \\
&= \frac{(1+\rho^2) - \rho(1+\rho^2)\sinh v}{1 - \rho^2 \sinh^2 v} = \frac{1+\rho^2}{1 + \rho \sinh v};
\end{aligned}
\tag{111}
$$

$$
v = \frac{2t}{T}\sqrt{1+\rho^2} + \sinh^{-1}(\rho);
\tag{112}
$$

$$
\sinh v = \rho \cosh\left(\frac{2t}{T}\sqrt{1+\rho^2}\right) + \sqrt{1+\rho^2}\sinh\left(\frac{2t}{T}\sqrt{1+\rho^2}\right).
\tag{113}
$$

This gives a final expression:

**Proposition 6.** *Consider the case when $H, p, \sigma^{-2} \to \infty$ in such a way that $H/p \to \infty$, while $H\sigma^2 \to \rho > 0$. Then for any $t, T > 0$,*

$$
\mathbb{E}[L(t)] \sim \frac{p}{2}\left(\frac{1+\rho^2}{1 + \rho^2 \cosh\psi + \rho\sqrt{1+\rho^2}\sinh\psi}\right)^2, \qquad \psi = \frac{2t}{T}\sqrt{1+\rho^2}.
\tag{114}
$$

For $\rho = 1$ (i.e. in the "conventional" mean-field limit), the above solution simplifies:

$$
\mathbb{E}[L(t)] \sim \frac{2p}{\left(1 + \cosh\left(\frac{\sqrt{8}t}{T}\right) + \sqrt{2}\sinh\left(\frac{\sqrt{8}t}{T}\right)\right)^2}.
\tag{115}
$$

For large $\rho$, we get $\mathbb{E}[L(t)] \sim \frac{p}{2}e^{\frac{-4\rho t}{T}}$, while as $\rho$ vanishes,

$$
\mathbb{E}[L(t)] \sim \frac{p}{2}(1 - \rho\sinh\psi)^2 \sim \frac{p}{2}\left(1 - 2\rho\sinh\left(\frac{2t}{T}\right)\right).
\tag{116}
$$

**Large time behavior.** As $t \to \infty$, we arrive at

$$
\mathbb{E}[L(t)] \sim 2p\left(\frac{1+\rho^2}{\rho^2 + \rho\sqrt{1+\rho^2}}\right)^2 e^{-\frac{4t}{T}\sqrt{1+\rho^2}}.
\tag{117}
$$

## E  EXPERIMENTAL DETAILS

**Infinite-data setting.** In our numerical experiments, instead of sampling a finite dataset $X$ and minimizing the empirical loss Eq. (2), we directly considered the infinite-data limit. Specifically, under the assumption that the data distribution has zero mean and identity covariance, we optimized the population loss

$$
L(U, V) = \frac{1}{2}\left\|U^\top V - I\right\|_F^2,
$$

which corresponds to the limit in Eq. (3). This allows us to isolate the intrinsic optimization dynamics of the factorization problem without additional stochastic effects due to finite sampling.

**Discretization and relation to gradient descent.** From the theoretical perspective, we analyze the continuous-time gradient flow dynamics Eq. (4). In numerical experiments, however, we implement standard gradient descent, which can be viewed as an explicit Euler discretization of the gradient flow. Concretely, for a discretization step $h > 0$, the updates take the form

$$
U_{k+1} = U_k - \frac{h}{T}\frac{\partial L(U_k, V_k)}{\partial U}, \qquad V_{k+1} = V_k - \frac{h}{T}\frac{\partial L(U_k, V_k)}{\partial V}.
$$

Thus, the effective learning rate of gradient descent is $\eta = h/T$. In the limit $h \to 0$, the discrete dynamics converge to the continuous gradient flow, while conversely, gradient flow provides an infinitesimal description of the training dynamics induced by gradient descent.

