# OpenReview forum: "Exact Learning Dynamics of a Linear Autoencoder Through Diagram Expansions"
_mathai.club/MathAI/2026/Conference — 2026 Oral_

### Official Review · Reviewer_tv1i · 2026-03-12
**Review for “Exact learning dynamics of a linear autoencoder through diagram expansions”**

**Rating:** 8
**Confidence:** 4

**Review:**

The paper studies the learning dynamics of a shallow linear autoencoder trained using gradient flow. The authors analyze the matrix factorization problem $I = W_2 W_1$ and derive analytic expressions for the evolution of the loss during training. The central methodological idea is to represent the learning dynamics through a diagrammatic expansion reminiscent of Feynman diagrams. This approach allows the authors to reformulate the analysis of gradient flow as a combinatorial problem on graphs. Using this framework, the paper derives explicit expressions for the average loss trajectory and analyzes the behavior of the system in different scaling regimes determined by the dimensions of the input and hidden layers. The authors introduce a geometric representation of these regimes via the notion of a \emph{Pareto polygon}. The theoretical results are supported by numerical experiments that demonstrate agreement between the analytical predictions and simulated training dynamics.

The paper addresses an important topic within the mathematical theory of machine learning, namely the analytical study of neural network learning dynamics. Exact analytical results of this type are relatively rare, even in simplified settings, and therefore the attempt to obtain closed-form descriptions of gradient dynamics is valuable.

The diagrammatic expansion proposed in the paper provides an interesting conceptual tool for organizing the perturbative expansion of learning dynamics. The work may be of interest to researchers studying the theoretical foundations of deep learning, particularly those working at the intersection of statistical physics, random matrix theory, and neural network theory.

The presentation is generally clear and follows a standard structure for conference papers. The experimental section confirms the analytical predictions and demonstrates that the derived formulas accurately describe the evolution of the loss in the considered setting.

The scope of the results is limited by the simplicity of the model considered. The analysis focuses on a shallow linear autoencoder, which is a highly simplified setting compared to modern neural network architectures. Although such models are commonly used as tractable theoretical testbeds, the paper would benefit from a clearer discussion of how the proposed approach could extend to more general settings, such as deeper linear networks or models with nonlinear activations.

In addition, some parts of the theoretical derivation are quite dense and may require careful reading. Additional intuition or illustrative figures explaining the diagrammatic construction could improve the accessibility of the paper.

The experimental evaluation is mainly illustrative and primarily serves to confirm the theoretical predictions. While this validation is useful, the experimental section remains relatively limited in scope.

The paper presents an interesting theoretical contribution to the study of neural network learning dynamics by introducing a diagrammatic expansion framework and applying it to a linear autoencoder model. The analytical results appear technically sound and the methodology is conceptually appealing. However, the broader impact of the work is somewhat constrained by the simplicity of the model and the limited discussion of possible extensions.

Overall, the paper could be a useful contribution to the conference audience, particularly for researchers interested in theoretical aspects of learning dynamics.

---

### Official Review · Reviewer_JLmG · 2026-03-12
**Promising conceptual idea lacking of practical support**

**Rating:** 6
**Confidence:** 2

**Review:**

The paper studies the learning dynamics of a linear autoencoder trained with gradient flow in the large-width and large-dimension regime. The authors develop a diagrammatic expansion framework to analytically compute the time evolution of the loss function. By representing terms of the loss and its derivatives as combinatorial graphs involving the model parameters, the paper derives series expansions for the expected loss under random Gaussian initialization. The authors analyze the asymptotic behavior of the dynamics and identify several regimes depending on the scaling of model width, input dimension, and initialization variance. These regimes include the neural tangent kernel (NTK) regime, mean-field regime, small initialization regime, and underparameterized nonlinear regime. The work provides analytic expressions for the expected loss evolution in several of these regimes and proposes a geometric interpretation of the scaling limits via a “Pareto polygon” that characterizes which diagrammatic contributions dominate in the large-scale limit.

**Strengths**

1. Novel analytical framework - the paper introduces a diagrammatic expansion approach for studying the dynamics of neural network training which is an interesting and relatively uncommon technique in the machine learning theory literature, and provides an alternative perspective to more standard analytical tools used in the analysis of neural networks.

2. Strong mathematical depth and technical development - the paper contains a substantial amount of technical derivations (e.g. combinatorial counting arguments and asymptotic analysis), the formalism is developed consistently and the derivations appear internally coherent, demonstrating a solid level of mathematical engagement with the problem.

3. Unification of different asymptotic regimes - the paper attempts to connect several well-known theoretical regimes within a single analytical framework. The proposed Pareto polygon interpretation provides an intuitive geometric picture of how different scaling assumptions lead to different dominant contributions in the diagrammatic expansion.

**Weaknesses**

1. Limited practical relevance of the model - the analysis is restricted to a single-layer linear autoencoder trained with gradient flow under Gaussian initialization and infinite-data assumptions. Such models are common in theoretical work, but the connection to modern ones is indirect, which limits the practical impact of the results.

2. Empirical validation is almost absent – the paper is almost entirely theoretical and does not include numerical experiments verifying the predicted learning dynamics. Simple simulations could have strengthened the paper by demonstrating that the theoretical regimes appear in finite-dimensional systems.

Overall, the paper presents an interesting theoretical approach to analyzing learning dynamics through diagrammatic expansions. The framework is mathematically rich and offer useful conceptual insights into the relationship between different asymptotic regimes of neural network training. However, the work focuses on a highly simplified model and lacks empirical validation, which limits its broader impact. The paper could be of interest primarily to researchers working on the theoretical foundations of deep learning, but its applicability to practical machine learning settings is limited.

---

> ### Author Rebuttal · Authors · 2026-03-13
>
> We thank the anonymous reviewer for their thorough evaluation, however, we would like to point out that the following "weakness" claim is false.
>
> *Empirical validation is almost absent – the paper is almost entirely theoretical and does not include numerical experiments verifying the predicted learning dynamics. Simple simulations could have strengthened the paper by demonstrating that the theoretical regimes appear in finite-dimensional systems.*
>
> **This is not true:** Figure 3 on page 7 of the main shows plots with numerical validation for all of the theoretical formulae we obtained. See also Appendix E for additional experimental details.

---

### Decision · Program_Chairs · 2026-03-14

**Decision:**

Accept (Oral)

**Comment:**

Dear Author(s),

On behalf of the Program Committee of the International Conference on Mathematics of Artificial Intelligence (MathAI 2026), we are pleased to inform you that your paper has been accepted for an oral presentation at MathAI 2026.

Your paper was evaluated through a rigorous two-stage review process involving both automated screening and expert review by members of the Program Committee. The reviewers recognized the quality and contribution of your work.

Presentation details:

- Format: Oral presentation (15–20 minutes + 5 minutes Q&A)
- Mode: You may present either in person (offline) at the conference venue in Sirius, Russia, or remotely via Zoom. Please indicate your preferred mode when confirming your participation.
- Conference dates: Marh 30 - April 3, 2026
- Website: https://mathai.club

Next steps:

1. Please confirm your participation and presentation mode by replying to this email mathai.club@yandex.ru no later than March 15, 2026 18:00 Moscow time.
2. If you plan to attend in person, the organizing committee will provide accommodation details separately.
3. Please prepare your final camera-ready manuscript according to the formatting guidelines available at https://mathai.club and upload it to OpenReview by March 15, 2026 18:00 Moscow time.

Should you have any questions regarding the program, logistics, or your presentation slot, please do not hesitate to contact us.

We look forward to your contribution to MathAI 2026.

With kind regards,

MathAI 2026 Program Committee
International Conference on Mathematics of Artificial Intelligence
https://mathai.club
OpenReview: https://openreview.net/group?id=mathai.club/MathAI/2026/Conference
Telegram: https://t.me/MathAI_club
Email: mathai.club@yandex.ru